# Relationship between Seasonal Changes in Food Intake and Energy Metabolism, Physical Activity, and Body Composition in Young Japanese Women

**DOI:** 10.3390/nu14030506

**Published:** 2022-01-24

**Authors:** Noriko Tanaka, Toyoko Okuda, Hisae Shinohara, Rie Shimonaka Yamasaki, Naomi Hirano, Jangmi Kang, Manami Ogawa, Nao Nishioka Nishi

**Affiliations:** 1Graduate School of Life Sciences, Kobe Women’s University, Kobe 654-8585, Hyogo, Japan; rieeee6@icloud.com (R.S.Y.); ogawa-nakano@suma.kobe-wu.ac.jp (M.O.); 2Faculty of Human Science, Tezukayama Gakuin University, Sakai 590-0113, Osaka, Japan; toyoko7992@yahoo.co.jp; 3Faculty of Education, University of Miyazaki, Miyazaki 889-2192, Miyazaki, Japan; e09102u@cc.miyazaki-u.ac.jp; 4Department of Food and Nutrition Sciences, Kobe Women’s Junior College, Kobe 650-0046, Hyogo, Japan; nhirano@kwjc.kobe-wu.ac.jp; 5Division of Nutrition Management, Heisei Medical Welfare Group, Japan & Department of Nutrition, Yodogawa Heisei Hospital, Osaka 533-0033, Osaka, Japan; kang.jangmi@hmw.gr.jp; 6Department of Arts and Science, Kobe Women’s Junior College, Kobe 650-0046, Hyogo, Japan; nnishioka@kwjc.kobe-wu.ac.jp

**Keywords:** seasonal change, energy intake, energy metabolism, body composition, physical activity, histidine, lean body mass, body fat

## Abstract

We investigated seasonal changes in food intake, energy metabolism, and physical activity (PA) and explored their associations with body composition. In total, 28 women aged 20–23 years in the Kansai area of Japan participated in this year-long study spanning the winter, spring, and summer seasons. A dietary investigation was performed using the weight recording method, and the amount of histidine in the diet, which may be related to the regulation of energy intake, was calculated. Resting metabolic rate (RMR), body composition, and PA were measured using indirect calorimetry, bioelectrical impedance analysis, and uniaxial accelerometry, respectively. The results showed that energy intake was highest in winter, decreased significantly with increasing temperature, and decreased by 25% in summer. As the intake of histidine in the diet did not increase in summer, it did not seem to be involved in the suppression of energy intake. RMR was highest in winter and decreased significantly in summer by 20%. The amount of PA was low in winter, increased significantly in the spring, and decreased again in summer. Body weight increased in winter, with an accumulation of fat in the trunk and arms, and decreased in summer, with a reduction in the amount of fat. Greater energy intake and less PA in winter induced an increment in body weight despite the increase in RMR. There were no significant changes in lean body mass between the seasons; however, the muscle weight of the lower limbs increased significantly in spring and in summer compared with that in winter (*p* < 0.001). Thus, seasonal changes in food intake, energy metabolism, and PA occur, with resultant changes in the body composition under comfortable air-conditioned environments.

## 1. Introduction

Food intake has long been known to be associated with climate, and spontaneous intake is low in areas with high environmental temperatures, such as deserts, but high in low-temperature areas, such as the poles [1]. In reality, many experience a loss of appetite in summer. In rats, food intake decreases, and water intake increases when animals are exposed to a hot environment for a long period of time, and this behavioral change results in the constant maintenance of body temperature [2,3]. Sakata et al. studied the physiological mechanism of suppression of food intake as an adaptive behavior. Using rats, they found that under a high-temperature environment of 31 °C, neuronal histamine synthesis in the hypothalamus increased, the satiety center was stimulated, and food intake decreased [4,5]. Moreover, a histidine decarboxylase inhibitor (FMH: α-fluoromethylhistidine), which inhibits neuronal histamine synthesis, was injected into the third cerebroventricle and the decrease in food intake was inhibited [4,6]. These results suggest that the decrease in food intake in a hot environment is caused by the enhancement of histamine synthesis. Histamine is synthesized from L-histidine [7], and it has been reported that histidine obtained from the diet serves as a source. However, there have been few studies on histidine levels obtained from the diet and the histamine-associated suppression of food intake in humans [8,9]. It is important to investigate the relationship between food intake and histidine in humans, as dietary surveys using dietary records report that most nutrients are obtained in winter more than in summer [10,11].

On the other hand, in terms of energy expenditure, the basal metabolic rate, which represents the minimum energy required for vital activities, is the energy expenditure common to all living organisms, and is the basis for calculating energy requirements. Climate-associated changes in the basal metabolic rate (BMR) and their relationship with physical activity (PA) have been assessed, and in Japan, where four distinct seasons and a clear temperature difference between summer and winter exists, it has long been observed that BMR increases in winter and decreases in summer [12,13]. In a study on seasonal variations in BMR, Oshiba reported that the basal metabolic rate in Japanese males increased by 6% when the air temperature decreased from 15 to 5 °C. In cold weather, metabolism is thought to increase to maintain the body temperature in order to adapt to the cold environment. However, in a subsequent study, Sasaki et al. reported that there has been a decline in fluctuation from 20% to <10% over the past 25 years in the observed annual variation (the extent of variation from the maximum value in winter to the minimum value in summer) in BMR [14,15]. More comfortable indoor environmental temperatures due to better air-conditioning equipment and changes in lifestyle and eating habits, such as an increase in fat intake, are mentioned as factors that may have led to the decline in fluctuation. Plasqui et al. also showed that the seasonal variation in sleep metabolism was lowest in summer (average temperature 21 °C) and highest in winter (average temperature 6 °C) [16]. Nevertheless, seasonal bias should be considered in the evaluation of energy intake from the viewpoint of energy balance in regions where basal metabolic rates differ between summer and winter.

Energy balance affects body weight and composition, and body fat thickness in particular within body composition is considered to be useful in thermoregulatory functions in response to changes in the climate. In other words, as the thermal conductivity of fat is lower than that of muscle, subcutaneous fat accumulation is considered effective for insulation in winter and, indeed, Ishigure et al. found that in female university students, regardless of whether they exercised regularly, there was a seasonal difference in subcutaneous fat thickness; the fat thickness increased in winter and decreased in summer [17]. On the other hand, Thais living in tropical climates have been reported to have a decreased sebum thickness compared to Japanese living in a temperate climate zone with four distinct seasons [18]. Recent studies have shown that both Thais and Japanese display seasonal variation in body fat percentage [19].

Food intake is thought to be associated not only with climate but also with mental and PA. Winter depression and seasonal affective (emotional) disorder, which are likely to occur in regions with short sunshine hours, reportedly increase appetite and carbohydrate intake in winter [20,21]. In addition, mental aspects are also related to PA, and depression is said to be low when PA is high [22]; however, Kaihara et al. reported that, even in a modern artificial living environment with reduced seasonal differences, seasonal changes in health status, mood, and behavior were observed in young individuals and they tended to become more active in summer with longer sunshine hours [23]. On the other hand, in terms of the relationship between food intake and PA, it was believed that food intake is increased to compensate for increased energy expenditure associated with higher PA [24]; however, recently, the relationship between food intake and energy metabolism through exercise has been reported to be weak [25].

Changes in BMR, food intake, body composition, and physical and mental health are thought to vary according to and interact with seasonal changes; however, few reports have simultaneously addressed these changes. Thus, in this study, food intake in the same younger female participants throughout winter, spring, and summer, particularly in winter and summer, including the relationship with histidine intake and food intake suppression, were evaluated and the seasonal changes in terms of energy metabolism, PA, and body composition were examined.

## 2. Materials and Methods

### 2.1. Target Participants

In total, 28 female students (including graduate students), aged 20 to 23 years, were recruited from Kobe University using posters, e-mail, and through contact with students in and around laboratories and lecture rooms. The participants lived in Kansai area, did not belong to an exercise club, and did not exercise regularly.

This study was approved by the Human Research Institutional Review Board of Kobe Women’s University (Approval no. H25-23) and was conducted only on those who provided consent after receiving an explanation about the purpose of the study, the method of data collection, etc.

### 2.2. Experimental Schedule

The investigation and testing in this study were performed between January and September 2014. The Japan Meteorological Agency [26] defines winter, spring, summer, and autumn as the 3 months of December to February, March to May, June to August, and September to November, respectively. In accordance with this, in this study, winter was defined as from January to February, spring from April to May, and summer from July to August. The average temperature, average humidity, and duration of sunshine for each period are shown in Table 1.

For 1–2 months in winter, spring, and summer, basal body temperatures were measured and recorded every morning and upon awakening to determine the low- and high-temperature periods, and the next investigation and measurement were performed in the low-temperature period, which is less affected by sex hormones. Body weight and composition were measured using bioimpedance, and diet was evaluated using a weight recording method for 3 consecutive days after the end of menstruation. In addition, resting metabolic rate was measured through exhaled gas measurements, and a questionnaire survey on the mind and body (The Center for Epidemiologic Studies Depression: CES-D) [25] was administered during this low-temperature period. The amount of PA was recorded and measured using a lifestyle recorder (Lifecorder Gs) that was worn daily on the waist for 1–2 months, the same period as that for basal body temperature measurements.

### 2.3. Measurement of Basal Body Temperature

Basal body temperature was measured using a women’s electronic thermometer (MC-642LC: OMRON). Measurements were taken and recorded sublingually every morning upon awakening. From the records of basal body temperature, the low-temperature period was defined as the period from the end of menstruation to the increase in basal body temperature and the high-temperature period was defined as the period between the increase in basal body temperature and immediately before it decreased.

### 2.4. Measurement of Resting Metabolic Rate (RMR)

Basal metabolic rate was measured immediately upon awakening after fasting for at least 12 h in the supine position, with the participant resting comfortably both mentally and physically. As the recruitment of participants into long-term experiments, such as this study, was expected to be difficult, RMR, which can be measured more easily than basal metabolic rate, was adopted. According to the Dietary Reference Intakes for Japanese, the RMR is calculated as 1.2 times BMR [27]. RMR was measured in 10 of 28 participants who provided informed consent, but data from only 7 participants were analyzed as 3 dropped out of the study early.

RMR was measured using a metabolic analyzer (MedGem^+^, MP Japan Co., Ltd., Tokyo, Japan) in an air-conditioned examination room. The room temperature was controlled to be approximately to 22–25 °C. After breakfast, participants fasted for at least 4 h, sat on a chair, and rested for at least 10 min prior to measurements.

### 2.5. Measurement of Physical Activity

The amount of PA was measured using an activity recorder based on a uniaxial accelerometry sensor (Lifecorder Gs: Suzuken Co., Ltd., Aichi, Japan). These were worn on the lower back every day, except during sleep and bathing. Days when these were forgotten to be worn were recorded on a form and excluded from the data. PA was measured from the first day of menstruation to the day before the next menstruation (1 cycle). Although there is individual variation, one cycle is about one month, and the daily averages of total energy expenditure, amount of exercise, number of steps, and METs·hour(H) were measured were calculated from the recorded data over one cycle.

Total energy expenditure (TEE) was calculated using the following formula:TEE (kcal/day) = basal metabolic rate + amount of exercise + amount of minimal motor activity + specific dynamic action (diet-induced thermogenesis: 10% of TEE)

BMR used in this analysis was calculated from each participant’s data, such as sex, age, height, and weight, based on the “Nutritional Requirements for Japanese: Basal metabolic rate per body surface area established in 1969” [28]. The amount of exercise was calculated from signals from the accelerometer, and the corresponding exercise coefficient was multiplied by body weight. The amount of minimal motor activity refers to energy expenditure during activities of very low intensity, such as having a conversation while standing.

### 2.6. Dietary Survey

Dietary surveys were conducted on three consecutive days during the low-temperature period using a self-administered weight-recording method [29]. The participants were asked to record the name of the menu item, the ingredients used in the menu item, the seasonings, and the food weight on a meal log. For the food weight measurements, all participants were given a digital kitchen scale (DKS-13: Kyowa Industrial Co., Ltd., Niigata, Japan) and practiced food weight measurement in advance. For commercially sold food products, the product name, distributor, and the amount of energy and nutrients listed were recorded. The participants were asked to take photographs of all food and beverages to prevent omissions in food records. The energy intake and nutrient intake per day were calculated using Excel Eiyou-kun version 6.0 (Kenpakusha, Tokyo, Japan).

### 2.7. Measurement of Body Weight and Body Composition

Body weight and body composition were measured using a body composition analyzer (Inbody720, Inbody Japan, Tokyo, Japan). Inbody720 is a multi-frequency bioimpedance analysis (BIA) that can independently measure body composition in 5 regions of each arm, each leg, and the trunk with an 8-point contact electrode on 6 different frequencies. Inbody720 has a high correlation with conventional methods, such as underwater weight measurement and dual-energy X-ray absorptiometry (DXA), with the values of the measured parameters and a high explanatory coefficient (R^2^) after linear regression analysis with DXA [30,31,32]. To improve the reliability of the measured values by BIA, the measurement was performed as follows. Body weight and body composition were measured for three consecutive days to monitor individual measurement and reliable data, immediately after the end of menstruation. The values of body composition were almost the same for all three times and the average values were used as data. Participants were required to have “completed urination and defecation”, “not have exercised before the measurement”, “not be wearing a pacemaker”, and “changed into a gown” “at least 3 h after breakfast”.

### 2.8. Questionnaire Survey on the Mind and Body

A questionnaire survey on the mind and body to evaluate the relationship between mental status and diet was administered. The Center for Epidemiologic Studies Depression (CES-D) (The National Institute of Mental Health) depression scale was used to assess the participants’ mental and physical state [33]. The CES-D is a questionnaire, wherein a total of 20 questions (0–3 points) need to be answered, with 16 negative and 4 positive questions, and rather than evaluating each item, the total score is calculated; a score of 16 or more indicates a tendency for depression [33].

### 2.9. Statistical Procedures

Statistical analyses were performed using SPSS (Statistics version 21, IBM, IL, Chicago, USA). Tests of differences between seasons employed repeated-measures ANOVA. The test of sphericity of Marchly was performed by one-way analysis of variance. If the sphericity could not be presumed, the degrees of freedom were corrected using the Greenhouse–Geisser ε. Subsequently, the Bonferroni multiple comparison test was performed. All values were expressed as the means ± SD (standard deviation). The level of significance was less than 5%.

For the sample size, the power of the test was detected using G*Power3 [34]. A sample size of 28 provided 80% power to detect the difference at a 2-tailed significance level of 0.05, on an effect size of 0.25. In RMR only (*n* = 7), however, the power was low: 20%.

## 3. Experimental Results

### 3.1. Participant Characteristics

The physique of the participants at the start of the investigation was as follows: height was 159.9 ± 3.9 cm, body weight was 54.0 ± 6.2 kg, and body mass index (BMI) was 21.1 kg/m^2^(Table 2), with height and weight slightly higher than and BMI equal to the national averages of women in their 20s as reported by the National Health and Nutrition Survey (2014) [35]: height, 157.8 ± 5.4 cm; weight, 52.5 ± 9.5kg; and BMI, 21.1. Moreover, among the participants, 3 (10.7%) were thin with a BMI < 18.5, 0 (0%) were overweight with a BMI ≥ 25 (obesity based on Japanese criteria), and 25 (89.3%) had BMI within normal ranges, demonstrating that many had physiques within normal ranges compared to the national average data of women in their 20s (thin: 17.4%; overweight: 10.4%; normal range: 72.2%).

### 3.2. Seasonal Changes in Energy and Nutrient Intake

Total food intake, which is the sum of water and food intakes, did not change significantly between seasons (Table 3). In addition, although water intake did not change between seasons, food intake significantly decreased from winter to summer (*p* = 0.011).

The energy intake in winter was 1782 kcal/day, which was the highest among the 3 seasons, and decreased significantly with increasing temperature in spring and summer, reaching 1331 ± 285 kcal/day in summer. The decrease in energy intake from winter to spring was 285 kcal and that from spring to summer was 166 kcal, showing that the decrease from winter to spring was greater. Indeed, energy intake decreased by 25% from winter to summer (Table 3).

The intake of protein and carbohydrates significantly decreased in summer compared to winter (prot: *p* = 0.05; CHO: *p* < 0.001). Fat intake decreased significantly in spring and summer when energy intake decreased. Therefore, seasonal changes in food intake were evident, with the highest intake of protein, fat, carbohydrates, and total energy occurring in winter, and the lowest intake of protein, fat, carbohydrates, and total energy occurring in summer.

There were no differences in the energy composition ratios of protein energy (P), fat energy (F), and carbohydrate energy (C), and no seasonal changes were observed.

Histidine intake per gram of protein (mg/g protein) was almost the same in all three seasons, and there were no significant seasonal changes in the amount of histidine per unit energy (mg/kcal) (Table 3).

### 3.3. Seasonal Changes in Physical Activity

Data on daily activity were used for the period from the first day of menstruation to the day before the next menstruation (1 cycle). One cycle was approximately one month of recorded data, and the amount of PA per day was calculated from the integrated data.

TEE calculated from the Lifecorder significantly increased from winter to spring, and then significantly decreased in summer (Table 4). The number of steps per day increased significantly from winter to spring, and then decreased significantly in summer. The number of steps was lowest in summer at 9298 ± 2570 steps and highest in spring at 10,922 ± 2523 steps.

The amount of exercise (METs·hour (H)) recorded using the Lifecorder was only calculated for exercises of ≥3 METs. The seasonal change in the METs·H was similar to that in the number of steps, with a significant decrease in winter, a significant increase in spring, and the maximum value seen, after which there is a subsequent decrease in summer and a return to levels similar to that in winter.

### 3.4. Seasonal Changes in RMR

Seven subjects agreed to participate in the measurement of their metabolism for RMR measurements and were able to provide data for the three seasons. The measured RMR was highest in winter and decreased in the spring and summer, particularly in summer, with a significant decrease of about 20% compared to that in winter (Table 4). This change was also similar for RMR per body weight and per lean body mass (LBM, same as fat-free mass (FFM)), decreasing from winter to spring and significantly decreasing in summer. Thus, RMR was clearly higher in winter and lower in summer, showing seasonal changes.

### 3.5. Seasonal Changes in Body Size and Composition

Body weight was higher in winter and significantly lower in summer than in spring (Table 2). The mean BMI of participants remained within the normal range of BMI between 18.5 and 25.0 throughout three seasons and was significantly lower in summer than in spring (*p* = 0.018).

Body fat mass was highest in winter when temperature was low, decreased in spring with increasing temperature, and decreased significantly in summer. Similarly, body fat percentage was highest in winter and tended to decrease with increasing temperature, with a particularly significant difference between winter and summer. On the other hand, LBM, and skeletal muscle mass showed no seasonal changes in winter, spring, and summer.

In terms of seasonal changes in body fat mass based on region, the fat mass in the left arm and trunk was highest in winter and lowest in summer; however, the fat mass in the right arm and in bilateral lower limbs did not change significantly.

Although there were no seasonal changes in LBM (same as FFM), muscle mass in the right and left upper limbs and trunk decreased from winter to spring and summer, particularly in summer, and decreased significantly compared to that in winter. On the other hand, the muscle mass of both lower limbs significantly increased in spring and summer compared to that in winter (*p* < 0.001) (Table 2).

### 3.6. Seasonal Changes in Depression

The CES-D depression scale was used to assess mental and physical status, but there were no significant differences between seasons (Table 5). A total score of 16 or higher on the 20-question item CES-D scale indicates a tendency toward depression. Among the participants of this study, the proportion with a total score of ≥16 was 40% to 50% in each season, and there were no significant differences.

## 4. Discussion

In this study, changes in food intake, energy metabolism, and body composition in young females with a normal BMI throughout three seasons (winter, spring, and summer) were investigated, and relationships with body composition, PA, and mental and physical status were evaluated.

The physique of participants at the start of the investigation was almost the same as the national average, which indicates a group representing the average physique among Japanese women.

### 4.1. Seasonal Changes in Food Intake

Food intake was highest in winter when the mean air temperature was 6.1–6.2 °C (minimum 3.0 °C; maximum 9.4 °C) and significantly decreased in spring and summer, respectively, with the lowest intake in summer when the mean air temperature was 27.3–27.4 ℃ (minimum 24.9 °C; maximum 30.6 °C), showing seasonal changes (Table 1). Energy intake varied similarly, with the highest being 1782 ± 436 kcal/day in winter and the lowest being 1331 ± 285 kcal/day in summer (Table 3). The energy intake in winter was higher than the average of 1662 ± 480 kcal/day among Japanese women in their 20s (2014 National Health and Nutrition Survey) [35]; however, values were lower than the average in other seasons. Thus, there may be a need to consider seasonal bias in evaluating results when conducting dietary surveys. The amount of drinking water consumed did not change in the three seasons, although it increased slightly in summer. The volume of water may appear to be small despite large standard deviation. The amount of water consumed by young females recorded in this study appears to vary widely among individuals.

The seasonal changes in food intake in this study are consistent with an old study that reported that spontaneous food intake was higher at the poles with lower environmental temperature and lower in deserts [1]. On the other hand, a dietary survey using FFQg [36] (a food frequency questionnaire based on food groups) administered to Japanese, Polish, and Thai individuals by Morinaka et al. did not show seasonal changes [19]. However, in a report from the Netherlands and Spain [10,11] using dietary records and a study using 24-h recall [37,38], food and energy intake in winter were clearly higher than those in summer, and the reduction in energy intake in summer was consistent with the results of the present study.

In terms of the decrease in energy intake in summer, Sakata et al. proposed in their animal study that the suppression of food intake under a high-temperature environment is due to an increase in neuronal histamine [4,5]. A negative correlation was also found between energy intake and histidine intake among those who ate fish, which is rich in histidine [8], suggesting that the intake of histidine-rich protein has a suppressive effect on food intake. When the dietary intake of histidine was calculated in this study, it was almost the same in winter, spring, and summer, with no significant seasonal changes in the amount of histidine per unit energy intake (Table 3). Thus, the speculation is that the decrease in food intake in summer was due to the enhanced histamine synthesis from histidine with increasing air temperature, rather than an increase in histidine intake from dietary protein.

Moreover, the intake of protein, fat, and carbohydrates, which constitute energy intake, decreased from winter to summer; however, there were no significant seasonal differences in P, F, and C ratios. These results suggest that the energy reduction in summer does not change the PFC ratio and decreases protein, fat, and carbohydrate uniformly. Typically, the Japanese conventional diet is a high-carbohydrate diet, which may facilitate the decrease in summer food intake. This is because a regular Japanese diet contains a carbohydrate energy ratio of 50–60%, which results in a larger volume of intake than a high-fat diet with the same energy, which causes satiety by serving as an afferent signal for the activation of hypothalamic histamine neurons by increasing the number of chews [39]. In addition, the higher temperature in summer activates hypothalamic histamine neurons. Thus, the bulkiness of Japanese foods, which are low in calories, may amplify the suppression of food intake under a high-temperature environment.

As for the relationship between food intake and PA, although there is a report that food intake increases with increasing PA [24], there is also a report that PA and food intake are not necessarily related [25]. In this study, the amount of exercise increased significantly from winter to spring and decreased significantly in summer (Table 4), whereas food intake decreased in the spring (Table 3) and did not correlate with changes in the amount of exercise. This result was consistent with the report that food intake does not necessarily increase with increasing PA [25].

In terms of seasonal changes for depression, there were no seasonal differences in the CES scores and frequency of scores ≥16 points according to the CES-D depression scale (Table 5). Kaihara et al. reported that moods are better and are more active in summer when the duration of sunshine is long [25]. In 2014, when this study was conducted, August had the highest precipitation in the previous 5 years; hence, a shorter duration of sunshine may have affected the finding that there were no seasonal differences in depression. There are also reports of higher depression and abnormal feeding behavior in winter when the duration of sunshine is short [20,24]. However, according to the data from this study, the duration of sunshine was not associated with depression, as the prevalence of participants with a score of ≥16 was constant in the range of 40–43% in any season.

### 4.2. Seasonal Changes in RMR and PA

RMR was highest in winter and decreased in spring and summer, and the change was significant particularly in summer (Table 4). This seasonal change was also observed per body weight and per LBM, with a decrease from winter to spring and a significant decrease in summer. Although energy metabolism is known to be affected by lean mass [40,41], the RMR per lean mass was also higher in winter and lower in summer. The rate of change in RMR in summer and winter was about 20% at a temperature difference of 20 °C. The rate of change in BMR was higher in the present study than that in an experimental study in young adult men by Oshiba et al., which reported that a 10 °C decrease in air temperature increased BMR by 6% [14]. Furthermore, Sasaki et al. reported that the annual variation range of BMR observed in the 25 years from 1949 to 1974 decreased from 20% to <10%, and one of the factors for this decrease was the improvement in function of air-conditioning equipment to make indoor environmental temperatures more comfortable [15]; however, even in the present age, the variation of RMR in winter and summer was approximately 20% in this study, showing no decrease in the variation. Thus, seasonal changes in energy metabolism may exist as an adaptation to environmental temperatures. Although no consensus has been reached among researchers regarding the presence or absence of seasonal variation in BMR, Plasqui et al. [16] found the largest and smallest seasonal variation in energy metabolism in winter and summer from measurements of the sleep metabolic rate, respectively; however, changes in energy metabolism could not be explained by changes in LBM, metabolism-related thyroid hormones, and leptin. In this study, increases in LBM and PA were not observed in winter, when energy metabolism increased. A previous study using the same experimental method as this study obtained a similar result, e.g., low in summer and high in winter of RMR [42]. In the present study, only 7 out of 28 subjects could measure RMR, which constituted a limitation of our study.

The amount of activity in winter is reportedly lower than that in other seasons [43] and similar results were obtained in this study; however, the amount of activity and number of steps increased significantly from winter to spring, decreased significantly from spring to summer, and reached the same levels as winter (Table 4). The frequency of going out may decrease in cold and hot environments, and such behavioral changes may be related to the small amount of exercise and the number of steps taken in winter and summer. In addition, the average number of steps taken per day by the participants of this study was more than 9000 throughout 1 year, which is much higher than the average number of 7028 steps taken by 20–29-year-old women according to a report from the National Health and Nutrition Survey in 2014 [35], showing that the amount of PA was higher than the general population. The METs·H (2.8 METs·H), which represents at least 3 METs activities, was 2.8 in winter, increased to 3.4 in spring, and decreased to 2.9 in summer, with the peak activity being in spring; however, the participants’ activities were around the level of the standard for Japanese PA and were considered appropriate. In addition, TEE significantly increased from winter to spring and decreased significantly in summer, the same as the seasonal changes in steps and amount of exercise (Table 4) but was not synchronized with changes in food intake or energy metabolism.

### 4.3. Relationship between Seasonal Changes in Food Intake and RMR, Body Weight, and Body Composition

Although energy intake decreased with increasing temperature and increased with decreasing temperature, showing seasonal changes (Table 3), the RMR of energy expenditure also showed higher values in winter and lower values in summer (Table 4). The rate of decrease in energy intake from winter to summer was about 25%, whereas that of energy expenditure was about 20%, suggesting that the energy balance would be negative as the decrease in energy intake would exceed the decrease in energy expenditure. Thus, for this reason, the body weight is thought to have decreased from winter to summer.

The decrease in body composition, fat mass, and fat percentage from winter to summer is shown in Table 2. The change in body fat percentage was consistent with that reported by Morinaka et al. [19], who conducted a survey in Japanese and Polish individuals and found a seasonal change in body fat percentage, which was lower in summer and higher in winter. In terms of changes in body fat mass based on region, the fat mass of the left arm and trunk decreased significantly from winter to summer, although the decrease in the mass of the right arm and leg was not significant. As the fat mass of the upper and lower limbs mainly consisted of subcutaneous fat, it was suggested that the decrease in lower limb fat in summer was a result of a decrease in subcutaneous fat. Subcutaneous fat has heat-insulating effects, and body fat increases at low temperatures. This is because subcutaneous fat accumulation is necessary to maintain body temperature [12], and body fat increases in winter but becomes unnecessary and may decrease in summer.

LBM (FFM) and skeletal muscle mass did not change in winter, spring, and summer, but muscle mass in the upper limbs and trunk decreased significantly in summer (Table 2). This decrease may be due to decreased food intake. On the other hand, the muscle mass in the lower limbs increased from winter to summer, unlike that in the trunk and upper limbs. The increase in muscle mass in the lower limbs in summer may be related to the increase in the amount of exercise in spring, and the increase in muscle mass may be delayed after the increase in the amount of exercise [44,45]. These results suggest that muscle mass changes according to activity rather than season.

## 5. Conclusions

Even in the present day, when the indoor environmental temperature becomes more comfortable, energy intake and energy metabolism changes seasonally, and the body composition changes with inactivity. Body fat mass was found to be higher in winter, when the temperature is low, whereas little fat mass was observed in response to summer climate. Thus, considering seasonal changes in energy intake, expenditure, and body composition, nutritional guidance and education may be necessary.

## Figures and Tables

**Table 1 nutrients-14-00506-t001:** Monthly average temperature, humidity, and daylight hours.

	Winter		Spring		Summer	
	Jan	Feb	Apr	May	Jul	Aug
temperature (℃)						
average	6.2	6.1	14.6	19.5	27.3	27.4
mini–max	3.0–9.7	3.4–9.4	10.8–18.8	16.0–23.6	24.9–30.6	25.3–30.2
humidity (%)	60	62	56	62	72	75
daylight hours	171.6	130.4	210.2	260.7	203.8	115.3

Data from Japan Meteorological Agency (JMA) in 2014. JMA http://www.jma.go.jp/jma/index.html (accessed on 22 September 2021).

**Table 2 nutrients-14-00506-t002:** Seasonal changes in body composition.

	Winter	Spring	Summer
Hours of sleep (min)	443 ± 49	428 ± 29	427 ± 61
Stature (cm)	159.9 ± 3.9		
Body Weight (kg)	54.0 ± 6.2	54.0 ± 5.9 ^c^	53.4 ± 6.1 ^c^
Fat mass (kg)	15.6 ± 4.5 ^b^	15.3 ± 4.4	14.9 ± 4.0 ^b^
Fat mass (%)	28.3 ± 5.5 ^b^	27.9 ± 5.5	27.4 ± 4.9 ^b^
Lean Body Mass (kg)	38.5 ± 3.0	38.8 ± 3.0	38.5 ± 3.4
Skeletal muscle mass (kg)	20.9 ± 1.8	21.0 ± 1.9	20.9 ± 2.0
BMI (kg/m^2^)	21.1 ± 2.1	21.1 ± 2.0 ^c^	20.9 ± 2.0 ^c^
R Arm Fat mass (kg)	1.04 ± 0.36	1.01 ± 0.35	0.99 ± 0.30
L Arm Fat mass (kg)	1.07 ± 0.36 ^b^	1.04 ± 0.34	1.01 ± 0.30 ^b^
Trunks Fat mass (kg)	7.29 ± 2.44 ^b^	7.08 ± 2.33 ^c^	6.79 ± 2.18 ^b, c^
R Leg Fat mass (kg)	2.60 ± 0.67	2.59 ± 0.66	2.55 ± 0.58
L Leg Fat mass (kg)	2.58 ± 0.69	2.58 ± 0.66	2.54 ± 0.58
R Arm Muscle mass (kg)	1.72 ± 0.22 ^b^	1.72 ± 0.21	1.69 ± 0.24 ^b^
L Arm Muscle mass (kg)	1.68 ± 0.21 ^b^	1.67 ± 0.21	1.65 ± 0.24 ^b^
Trunks Muscle mass (kg)	16.65 ± 1.41 ^b^	16.60 ± 1.34	16.41 ± 1.55 ^b^
R Leg Muscle mass (kg)	6.05 ± 0.64 ^a, b^	6.16 ± 0.68 ^a^	6.19 ± 0.67 ^b^
L Leg Muscle mass (kg)	6.03 ± 0.65 ^a, b^	6.14 ± 0.67 ^a^	6.18 ± 0.69 ^b^
	Means ± S.D. (*n* = 28)		

R, Right; L, Left; Lean Body Mass(LBM) is the same as Fat-Free Mass(FFM); Bonferroni, ^a, b, c^ Means with a common superscript letter are significantly different, *p* < 0.05.

**Table 3 nutrients-14-00506-t003:** Seasonal changes in food weight, energy, and nutrient intakes.

	Winter	Spring	Summer
TFW (g)	1731 ± 516	1642 ± 394	1580 ± 485
Amount of drinking (g)	655 ± 343	638 ± 333	693 ± 298
Food weight (g)	1076 ± 306 ^b^	1004 ± 248	886 ± 266 ^b^
TEI (kcal)	1782 ± 436 ^a, b^	1497 ± 335 ^a, c^	1331 ± 285 ^b, c^
Protein (g)	62.7 ± 19.0 ^b^	54.2 ± 15.3	48.4 ± 14.0 ^b^
Fat (g)	63.1 ± 17.8 ^a, b^	50.4 ± 14.9 ^a^	44.2 ± 16.4 ^b^
Carbohydrate (g)	232.0 ± 64.6 ^b^	199.6 ± 47.3	179.7 ± 39.0 ^b^
P ratio (%)	14.0 ± 2.8	14.5 ± 2.5	14.5 ± 2.6
F ratio (%)	31.2 ± 5.9	29.9 ± 4.5	28.2 ± 8.5
C ratio (%)	52.7 ± 7.3	53.7 ± 5.6	54.9 ± 8.3
Histidine (mg)	1659 ± 587 ^b^	1456 ± 488	1301 ± 496 ^b^
Histidine (mg/g protein)	26.2 ± 3.3	26.8 ± 3.4	26.4 ± 4.6
Histidine (mg/kcal)	0.92 ± 0.24	0.98 ± 0.23	0.97 ± 0.29
	Means ± S.D. (*n* = 28)		

TFW, Total food weight including drinks; TEI, Total energy intake; P ratio, protein energy ratio, percentage of energy intake from protein; F ratio, fat energy ratio, percentage of energy intake from fat; C ratio, carbohydrate energy ratio, percentage of energy intake from catbohydrarte; Bonferroni, ^a, b, c^ Means with a common superscript letter are significantly different, *p* < 0.05.

**Table 4 nutrients-14-00506-t004:** Seasonal changes in physical activity and resting metabolic rate (RMR).

	Winter	Spring	Summer
TEE (kcal/day)	1796 ± 152 ^a^	1843 ± 160 ^a, c^	1782 ± 160 ^c^
PAEE (kcal/day)	230 ± 67 ^a^	274 ± 69 ^a, c^	233 ± 70 ^c^
Steps	9359 ± 2473 ^a^	10922 ± 2523 ^a, c^	9298 ± 2570 ^c^
METs·Hour	2.8 ± 0.9 ^a^	3.4 ± 0.8 ^a, c^	2.9 ± 1.0 ^c^
RMR (kcal)	1240 ± 142 ^b^	1131 ± 149	986 ± 111 ^b^
RMR (kcal/kg BW)	24 ± 3 ^b^	22 ± 2	19 ± 2 ^b^
RMR (kcal/kg LBM)	33 ± 3 ^b^	30 ± 3	26 ± 2 ^b^
	Means ± S.D. (*n* = 28)		

TEE, Total Energy Expenditure; PAEE, Physical-activity-related Energy Expenditure by a uniaxial accelerometer; RMR, Resting Metabolic Rate (*n* = 7); LBM, Lean Body Mass; BW, body weight; Bonferroni, ^a, b, c^ Means with a common superscript letter are significantly different, *p* < 0.05.

**Table 5 nutrients-14-00506-t005:** Seasonal changes in the total CES-D score and percentage of subjects with CES ≥ 16 points.

	Winter	Spring	Summer
CES Total score	13.2 ± 7.5	14.6 ± 8.6	13.3 ± 8.5
CES ≥ 16 points (%)	11 (39.2%)	12 (42.9%)	11 (39.2%)
	Means ± S.D. (*n* = 28)		

CES-D, The Center for Epidemiologic Studies Depression; ≥ 16 points(%), Percentage of subjects with a score of 16 points or more.

## Data Availability

The data presented in this study are available on request from the corresponding author.

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
