# Peer review of "Relationship between Seasonal Changes in Food Intake and Energy Metabolism, Physical Activity, and Body Composition in Young Japanese Women"

_nutrients, 2022, doi:10.3390/nu14030506_

Round 1

Reviewer 1 Report

Line 67 "increases the calorific value" ? Calorific value of what? Also calorific value of something usually refers to its inherent ability to generate energy. Please check this sentence and rephrase.

Line 68. I am not sure of the point being made here. I assume that you mean there has been in a decline over the past 20 years in the observed annual variation in BMR. The present wording is a little unclear, please consider rephrasing.

Line 73 "decrease in fluctuation" I assume you mean in magnitude not the periodicity of the fluctuation. The sentence is potentially ambiguous - suggest rephrasing.

Line 78 on. A careful distinction needs to be made here. "fat" is a chemical entity which exists in the body predominantly in "adipose tissue" to which you are referring here. Please make this clear, for example, "fat thickness" really relates to "adipose tissue thickness". 

Methods 

Where were the studies conducted? Awakening temperature could be done at home but BMR (RMR) on awakening was presumably in a metabolic ward. Please provide details of the study setting.

It appears that only complete data for 7 participants were obtained (Line 157). If this is correct this study is very small and is most likely underpowered. Please provide power calculations and sample size need to meet the study aims. Also this raises the question of the sample size for data presented in the results. Is this for the 7 completers only or a variable number for the different measurements? If the former, then sample size becomes critical, if the latter then data are not comparable. This is a key problem with the study.

Line 171 onward. The calculation of TEE is unclear for example, how was SDA calculated? 

Section 2.6 I assume that this was self-conducted at home. Please confirm. If so do you have reliability data for the survey.

Line 195. What do you mean by "types of frequencies"? In bioimpedance there is simply the frequency of the applied AC current.

Line 196. A high correlation does not mean high agreement. You could have perfect correlation but very poor agreement.  How were these measurements performed? BIA is an indirect method that requires a high degree of standardisation for accuracy. Provide details.

Line 199. The mean of 3 values was used but what if they were very different? Was this ignored? What was the daily variation?

Table 2. How many participants for these data? Also I query the data. Take Winter as an example. Mean body weight was 54 kg. FM was 15.6 kg. Lean was 38.5 kg, i.e. lean + FM = 54.1 kg. BIA is a 2C model, i.e. BW = FM + FFM. By comparison, DXA is a 3C model BW = FM + lean + BMC. For your data "Lean" would actually appear to be FFM or the participants had no bone mineral (skeleton), clearly not plausible. Please check your results and get the terminology correct. I suspect that InBody are using the term "lean" incorrectly actually meaning FFM.

In addition, I am not clear about the muscle values. Again for the Winter, the sum of the segment muscle masses = (1.72 + 1.68 + 16.65+ 6.05 + 6.03) = 32.1 kg This is way above the skeletal muscle mass and approaching the total lean mass. This cannot be correct. There cannot be 16.65 kg of MUSCLE in the trunk. Again I think that this is confusion of terminology. Please check your data carefully.

Table 3. Were these participants really only having a total of around 650 mL of liquid per day? This seems very low. Even in summer when temperatures were around 27C and 75% humidity this rose only to 698 mL.  Valtin in his review estimates basal adult water requirement at around 1700 mL per day https://doi: 10.1152/ajpregu.00365.2002. Even if we allow for metabolic water this "fluid" intakes seem unusually low. See https://doi.org/10.1038/ejcn.2014.290 Again check and comment please.

The Discussion may require a re-write in the light of comments above regarding the data and sample number

Author Response

Dear R1 reviewer,

Thank you so much for your valuable comments concerning our manuscript entitled  “ Relationship between seasonal changes in food intake and energy metabolism, physical activity, and body composition ”.

We have studied the comments carefully and have made corrections, according to the comments, which I hope meet with their approval.

Q1. Line 67 "increases the calorific value" ? Calorific value of what? Also calorific value of something usually refers to its inherent ability to generate energy. Please check this sentence and rephrase

A1. The word “the caloric value” might be confusing meaning and so I rewrote this sentence.

L 69-70

Q2. Line 68. I am not sure of the point being made here. I assume that you mean there has been in a decline over the past 20 years in the observed annual variation in BMR. The present wording is a little unclear, please consider rephrasing.

A2. Thank you.  I rewrote this sentence.  L70-73

Q3. Line 73 "decrease in fluctuation" I assume you mean in magnitude not the periodicity of the fluctuation. The sentence is potentially ambiguous - suggest rephrasing.

A3. Yes. It means in magnitude of fluctuation.  I added a sentence.  L70-73

Q4. A careful distinction needs to be made here. "fat" is a chemical entity which exists in the body predominantly in "adipose tissue" to which you are referring here. Please make this clear, for example, "fat thickness" really relates to "adipose tissue thickness".   Line 78

A4. Thank you for your suggestion. I changed to “fat thickness”. L81

Methods

Q5. Where were the studies conducted? Awakening temperature could be done at home but BMR (RMR) on awakening was presumably in a metabolic ward. Please provide details of the study setting.

A5.  RMR was measured in an air-conditioned examination.  I added the sentences. L169-170

Q6. It appears that only complete data for 7 participants were obtained (Line 157). If this is correct this study is very small and is most likely underpowered. Please provide power calculations and sample size need to meet the study aims. Also this raises the question of the sample size for data presented in the results. Is this for the 7 completers only or a variable number for the different measurements? If the former, then sample size becomes critical, if the latter then data are not comparable. This is a key problem with the study.

A6. Most measurements such as body composition, dietary survey, physical activity were collected from 28 subjects but only RMR was 7 measurements. It was 7 subjects who agreed to measure RMR through 3 seasons. The sample size of 7 was small and so, I added limitations of our study in Discussion. L165-167, L 488-490

Q7. Line 171 onward. The calculation of TEE is unclear for example, how was SDA calculated? 

A7. SDA (DIT) is included in TEE as 10% of TEE.    L182-183

Q8. Section 2.6 I assume that this was self-conducted at home. Please confirm. If so do you have reliability data for the survey.

A8. The participants practiced digital kitchen scale which we gave in advance and were asked to take photographs of all food and beverages to prevent omissions in food records. L197-200

Q9 Line 195. What do you mean by "types of frequencies"? In bioimpedance there is simply the frequency of the applied AC current.

A9. Inbody 720 is multi-frequency BIA with eight-point tactile electrode. I rewrote this sentence.  L204-207

Q10. Line 199. The mean of 3 values was used but what if they were very different? Was this ignored? What was the daily variation?

A10.  We measured body composition for 3 consecutive days to monitor the same person's unusual measurements, but body composition was always measured at constant times and conditions, and the data for the three times were almost the same.  L 210-217

Q11. Table 2. How many participants for these data? Also I query the data. Take Winter as an example. Mean body weight was 54 kg. FM was 15.6 kg. Lean was 38.5 kg, i.e. lean + FM = 54.1 kg. BIA is a 2C model, i.e. BW = FM + FFM. By comparison, DXA is a 3C model BW = FM + lean + BMC. For your data "Lean" would actually appear to be FFM or the participants had no bone mineral (skeleton), clearly not plausible. Please check your results and get the terminology correct. I suspect that InBody are using the term "lean" incorrectly actually meaning FFM.

A11. LBM (lean body mass) is the same as FFM, which is calculated by subtracting body fat from body weight.  References that use “LBM”, Obesity, 2014, 22(6) 1546-52

 Q12. In addition, I am not clear about the muscle values. Again for the Winter, the sum of the segment muscle masses = (1.72 + 1.68 + 16.65+ 6.05 + 6.03) = 32.1 kg This is way above the skeletal muscle mass and approaching the total lean mass. This cannot be correct. There cannot be 16.65 kg of MUSCLE in the trunk. Again I think that this is confusion of terminology. Please check your data carefully.

A12. We did not put muscle data in this study, because muscle and skeletal muscle were confusing.

Q13. Table 3. Were these participants really only having a total of around 650 mL of liquid per day? This seems very low. Even in summer when temperatures were around 27C and 75% humidity this rose only to 698 mL.  Valtin in his review estimates basal adult water requirement at around 1700 mL per day https://doi: 10.1152/ajpregu.00365.2002. Even if we allow for metabolic water this "fluid" intakes seem unusually low. See https://doi.org/10.1038/ejcn.2014.290 Again check and comment please.

A13.Regarding volume of round 655±343 mL of liquid per day, the volume seems very low as compared with 1700ml in Valtin’s review.  But in survey of water intake, volume of Japanese women aged 48.6 years(average) was 934-998ml (average), approaching our data in spite of the standard deviation (in European Journal of clinical nutrition, 2015, 69, 907-913).  The amount of water consumed by young women recorded in this study appears to vary widely among individuals. Although I did not focus on amount of water intake, I added some sentences.  L413-416

We have made corrections, according to the comments, which I hope meet with their approval. Thank you. 

Reviewer 2 Report

This is an interesting piece of work on long-term changes in intake, physical activity, and body composition.

The aim of this study is to evaluated food intake in the same younger female participants throughout winter, spring, and summer, particularly in winter and summer, including the relationship with histidine intake and food intake suppression, were evaluated and the seasonal changes in terms of energy metabolism, PA, and body composition were examined.

The title of the work conforms to the content of this, is informative and concise. Although the work focuses on histidine

The sample size is 28 women, this is a very small sample size, therefore I consider it to be a pilot study, and this should be reflected in the title.

They used a dietary investigation was performed using the weight recording method, and the amount of histidine in the diet. Histidine is a nutritionally essential amino acid with many recognized benefits to human health, while circulating concentrations of histidine decline in pathologic conditions. According to the FAO, the daily requirement for histidine is 8 to 12 mg / kg of body weight per day in adults.

They concluded that, seasonal changes in food intake, energy metabolism, and PA occur, with resultant changes in the body composition under comfortable air-conditioned environments.

Comments:

The introduction should be based on human studies. I do not understand using experimental studies in rats. Since, as the authors indicate, the intake depends not only on the weather but also on psychology.

I suggest they rewrite the introduction and use human literature.

Material and methods

the sample size should be justified by an adequate calculation based on the objective of the study; I suggest adding the sample size calculation.

It would be interesting to know how the participants were recruited and their long-term participation rate.

Reference should be made in the bibliography to " Nutritional Requirements for Japanese: Basal metabolic rate per body surface area established in 1969 "(line 174-175).

Which dietary survey do you used? Please indicate if it is a standardized survey and the corresponding bibliographic reference.

Statistical analysis should be more fully explained.

Results

The tables are informative. the results being well structured.

Discussion

On line 368-369 it seems that we need to finish writing something, please check it.

The same occurs as in the introduction, bibliography should be used in humans, not in animals.

study limitations need to be identified

The conclusions are adequate since they correspond with the results.

Author Response

Nutrients-1520869

Dear R2 reviewer,

Thank you so much for your valuable comments concerning our manuscript entitled  “ Relationship between seasonal changes in food intake and energy metabolism, physical activity, and body composition ”.

We have studied the comments carefully and have made corrections, according to the comments, which I hope meet with their approval.

Q1. The introduction should be based on human studies. I do not understand using experimental studies in rats. Since, as the authors indicate, the intake depends not only on the weather but also on psychology.

I suggest they rewrite the introduction and use human literature.

A1. The introduction is based on human studies in many cases except research study of food intake regulation of histidine. Regarding animal studies, as the mechanism of food intake regulation has been obtained only by animal experiments, this time we investigated whether histidine intake in the nutrition survey is involved in the suppression of food intake in summer.  We added further information on seasonal changes in human nutrition surveys in introduction.  L57-59

Material and methods

Q2. the sample size should be justified by an adequate calculation based on the objective of the study; I suggest adding the sample size calculation.

A2. We calculated sample size of 28 to detect 81% power. In only RMR(n=7) , however, the sample size was small and added limitation to our study in discussion. L233-235, L488-490

Q3. It would be interesting to know how the participants were recruited and their long-term participation rate.

A3. We added the way the participants were recruited for the research program. L113-116

Q4. Reference should be made in the bibliography to " Nutritional Requirements for Japanese: Basal metabolic rate per body surface area established in 1969 "(line 174-175).

Which dietary survey do you used? Please indicate if it is a standardized survey and the corresponding bibliographic reference.

A4. I added the bibliography to " Nutritional Requirements for Japanese: Basal metabolic rate per body surface area established in 1969 "  L186 ref (28)

Q5. Which dietary survey do you used? Please indicate if it is a standardized survey and the corresponding bibliographic reference.

A5. We added the reference.  L193 Ref(29)

Q6. Statistical analysis should be more fully explained.

A6. I rewrote the sentences. L227-232

Results

The tables are informative. the results being well structured.

Thank you.

Discussion

Q7. On line 368-369 it seems that we need to finish writing something, please check it.

A7. Sorry. The sentence was cut off and went to the next paragraph. I corrected the sentense. L396-399

Q8. The same occurs as in the introduction, bibliography should be used in humans, not in animals.

A8. We put bibliography of humans in introduction.

Q9. study limitations need to be identified

A9. We added limitations to the study. L488-490

The conclusions are adequate since they correspond with the results.

Thank you.

We have made corrections, according to the comments, which I hope meet with their approval.  Thank you.

Round 2

Reviewer 2 Report

I have reviewed with interest the new version of the manuscript entitled " Relationship between seasonal changes in food intake and energy metabolism, physical activity, and body composition ", Nutrients-1520869.

I have verified that the authors have followed the suggestions made, so that the manuscript has improved from my point of view.

However, I think the title should fit more closely with the content as I indicated in the previous review.

Author Response

Dear R2 reviewer,

Thank you for your valuable comment.  Following  reviewer's suggestion,  I changed to the title  " Relationship between seasonal changes in food intake and energy metabolim, physical activity, and body composition in young Japanese women."  I hope meet with an approval.   Thank you. 
